# The Safety and Outcome of Minimally Invasive Staged Segmental Artery Coil Embolization (MIS^2^ACE) Prior Thoracoabdominal Aortic Aneurysm Repair: A Single-Center Study, Systematic Review, and Meta-Analysis

**DOI:** 10.3390/jcm13051408

**Published:** 2024-02-29

**Authors:** Vaiva Dabravolskaite, Eleni Xourgia, Drosos Kotelis, Vladimir Makaloski

**Affiliations:** 1Department of Vascular Surgery, Inselspital, Bern University Hospital, University of Bern, Freiburgstrasse 18, 3010 Bern, Switzerland; vaivadabravolskaite@gmail.com (V.D.); elena.xourgia@gmail.com (E.X.); drosos.kotetlis@insel.ch (D.K.); 2Department of Vascular Surgery, Turku University Hospital, 20100 Turku, Finland; 3Satasairaala Hospital, 28100 Pori, Finland

**Keywords:** segmental artery, coil embolization, thoraco-abdominal aortic aneurysm repair

## Abstract

Background: Minimally Invasive Staged Segmental Artery Coil Embolization (MIS^2^ACE) is a novel technique of spinal cord preconditioning used to reduce the risk of paraplegia in thoracoabdominal aortic aneurysm (TAAA) repair. In this study, we report our experience with MIS^2^ACE, including both degenerative and post-dissection TAAA, while we attempt to systematically summarize relevant data available in the literature. Design: single-center observational study with systematic review of the literature and meta-analysis. Methods: Initial retrospective analysis of 7 patients undergoing MIS^2^ACE over 12 sessions with a subsequent systematic review of the literature and meta-analysis of the available published data (PROSPERO protocol number: CRD42023477411). Baseline patient and aneurysm characteristics, along with procedural technique and outcomes, were analyzed. One-arm pooling of proportions was used to summarize available published data. Results: We treated seven patients (5 males, 71%) with a median age of 69 years (IQR 55,69). According to the Crawford classification, five patients (1%) had extent II TAAA, and two (29%) had extent III TAAA. Five patients (71%) had post-dissection -TAAA; four of them were after Stanford type A dissection, and one had a chronic type B dissection. Three patients (43%) had connective tissue disease. Of the seven patients, six (86%) underwent previous aortic surgery, while the median aneurysm diameter was 58 mm (IQR 55,58). MIS^2^ACE was successful in 11 sessions (92%). The median number of embolized arteries was 4 (IQR 1,4). There were no periprocedural complications in any embolization. The median embolization-operation time interval was 37.0 days (IQR 31,78). Two patients had open and five endovascular treatment. There were no events of spinal cord ischemia either after MIS^2^ACE or after the aortic repair. Out of the 432 initially retrieved articles, we included two studies in the meta-analysis, including patients with MIS^2^ACE for spinal cord preconditioning in addition to our cohort. The prevalence of pooled postoperative spinal cord ischemia among MIS^2^ACE patients is 1.9% (95% CI −0.028 to 0.066, *p* = 0.279; 3 studies; 81 patients, 127 coiling sessions). Conclusions: While the current published data is limited, our study further confirms that MIS^2^ACE is a technically feasible and safe option for spinal cord preconditioning.

## 1. Introduction

Spinal cord ischemia (SCI) remains to be the most devastating complication after open or endovascular treatment for thoracoabdominal aortic aneurysm (TAAA). Depending on the extension of underlying aortic pathology, the risk of SCI varies between 8% and 17% for open thoracoabdominal aortic repair and 2% and 17% for total endovascular aortic repair [1,2]. Different perioperative protocols proved to decrease the incidence of SCI. In particular, cerebrospinal fluid drainage (CSFD), a staged repair of TAAAs, and the preoperative preconditioning of the spinal collateral area with minimally invasive staged segmental artery coil embolization (MIS^2^ACE) in an elective setting was shown to lead to a better neurologic outcome [3,4]. The efficacy and safety of MIS^2^ACE have been previously proven in various experimental settings, including animal models, and in 2014, it was introduced in clinical practice as a novel technique of spinal cord preconditioning prior to thoracoabdominal aortic aneurysm (TAAA) repair [5,6]. Currently, MIS^2^ACE is being investigated in an international, randomized, multicenter control trial PAPAartis [7].

In this study, we report the safety and outcome after MIS^2^ACE, including both degenerative and post-dissection TAAA, while we attempt to systematically summarize relevant data available in the literature.

## 2. Methods

The present work consists of two components: an observational study and a subsequent systematic review of the literature and a meta-analysis of proportions in which the results of the observational study were combined with the results of relevant studies.

### 2.1. Observational Study

#### 2.1.1. Study Design

We conducted an observational retrospective cohort study including all consecutive patients undergoing MIS^2^ACE as preparation for an open or endovascular TAAA repair in a tertiary center (Inselspital, Bern, Switzerland) from January 2021 to September 2023. We followed the “Strengthening the Reporting of Observational Studies in Epidemiology” (STROBE) statement guidelines.

#### 2.1.2. Patient Group and Data Collection

MIS^2^ACE was performed in a selective group of patients requiring open or endovascular TAAA repair with multiple patent segmental arteries, both from the true and false lumen in patients with a dissected aorta, and deemed with higher risk for SCI during the following aortic repair. All indications were discussed at the house-intern aortic board, consisting of the leading vascular and cardiac surgeons and radiologists. The segmental arteries occlusion as a preemptive procedure was offered only to patients prior to elective aortic repair and being aware of the potential periprocedural risks. Access to the segmental arteries was gained via a transfemoral retrograde approach, under local anesthesia, to be able to monitor periprocedural neurological status and complications. A team of two vascular surgeons performed all the procedures. The operator selected appropriate material based on aortic anatomic characteristics and a planned approach. A maximum of six segmental arteries were occluded per session. According to the number of segment arteries intended to occlude, the number of sessions varied between one and three. Patients who had more than one MIS^2^ACE were calculated additionally as per session in the observational study.

The procedure is performed in either an angio suite or a hybrid operating theater and under local anesthesia, which ensures continuous monitoring of the patient’s neurological function. A preoperative CT angiography is initially performed as part of the procedure planning, which allows for a more precise evaluation of the patient’s anatomy prior to the segmental artery catheterization. We introduced a 60 cm long 6-Fr angulated sheath over a unilateral femoral access. This is followed by a five French catheter, which facilitates entrance to the target vessel’s ostium, followed by a microcatheter and guidewire for stable access to the main stem of the targeted segmental artery. Intraoperative digital subtraction angiography (DSA) followed in multiple projections to identify the segmental arteries. A selective segmental artery angiography is made to measure the size and the length of the segmental artery that has to be occluded. An appropriate size of detachable coils is chosen and introduced through the microcatheter. To ensure the safety of the procedure, a thorough neurological assessment is performed after each segmental artery coiling. In case of back pain, the procedure was interrupted immediately, as it was considered a sign of possible ischemia. No spinal fluid drainage catheter was inserted prior to MIS^2^ACE.

After each session, a postoperative close monitoring period of at least 48 h was followed in the surgical intermediate care unit. Technical success was defined as successful coil artery embolization of at least one segmental artery within the target aortic segment planned for future coverage. A focused sensorimotor neurological examination was performed every four hours during that immediate postoperative follow-up period. Vital parameters were continuously monitored. All patients were followed up clinically and radiographically for at least three months after the intervention. CT-angiographies were retrospectively reviewed by an author who was not part of the MIS^2^ACE operative team. We gathered data on demographics, aneurysm characteristics, interventional technical details, perioperative complications, and postoperative outcomes from a review of electronic patient records in an anonymized database for subsequent analysis. The extent of the aneurysm was classified according to the Crawford Classification. For post-dissection TAAA, initial dissection was classified according to the Stanford Classification. Open or endovascular treatment in one or several stages followed.

#### 2.1.3. Outcomes

We considered spinal cord ischemia within 48 h after MIS^2^ACE as the primary outcome of the observational study. Spinal cord ischemia at seven days, technical success of MIS^2^ACE, perioperative major bleeding, acute renal failure at 30 days, and all-cause mortality at 30 days were the secondary outcomes.

### 2.2. Meta-Analysis

#### 2.2.1. Study Design

After summarizing the data of our cohort, we carried out a systematic review and meta-analysis of relevant studies in accordance with the Preferred Reporting Items for Systematic Reviews and Meta-Analyses (PRISMA) statement. We prespecified search strategy, data extraction, and outcomes in a protocol registered with PROSPERO (CRD42023477411) and available online. The Institutional Review Board waived the need for ethical approval due to the retrospective nature of the study and the limited number of patients.

#### 2.2.2. Search Strategy

Two authors (VD and EX) independently conducted the literature search. In addition to PubMed and CENTRAL, we systematically searched preprint servers (namely medRxiv and Research Square) to capture rapidly accumulated evidence. We used Boolean logic to create the search phrase: (“MIS2ACE” OR “MIS²ACE” OR “segmental artery coil*”) OR (“stage*” AND “thoracoabdominal” AND “repair”). We retrieved relevant literature up to 27th November 2023, with no language restrictions. Studies reporting data on MIS2ACE and presenting patient mortality and/or morbidity were considered for inclusion. Case reports and case series involving less than five patients were excluded.

#### 2.2.3. Data Extraction and Risk of Bias Assessment

Two authors (VD and EX) independently extracted data from a prespecified worksheet and cross-checked their findings. For each included study, we collected data on the author, country, study design, number of patients undergoing MIS^2^ACE, patient and intervention characteristics (i.e., demographics, aneurysm size, and classification, number of arteries coiled), and outcomes. Two authors (VD and EX) independently assessed the risk of bias of included studies. Any disagreements were discussed with the corresponding author (VM). We used the Tool to Assess Risk of Bias in Cohort Studies to assess the included studies developed by the CLARITY Group at McMaster University.

#### 2.2.4. Outcomes

We considered spinal cord ischemia after MIS^2^ACE and after the aortic repair as the primary outcome of the meta-analysis.

### 2.3. Statistical Analyses

For the observational study, we used SPSS software 22.0 (IBM, Armonk, NY, USA). We presented continuous variables as the median and interquartile range (IQR). We presented categorical variables as the number of patients (percentage). For the single-arm meta-analysis of proportions, we used the metafor package of R. The pooling of proportions was carried out with a random effects model using the DerSimonian and Laird method, and results were presented as proportions with 95% confidence intervals.

## 3. Results

### 3.1. Observational Study

During the study period, seven patients [28.5% female, median age 57.0 (IQR 55.0–69.0) years] underwent MIS^2^ACE over 12 sessions in our clinic. Table 1 shows the baseline characteristics of perioperative technical details and outcomes of included patients.

### 3.2. Underlying Aortic Pathology

According to the Crawford classification, five patients (71%) had extent II TAAA and two (29%) extent III TAAA. Five patients (71%) had post-dissection TAAA, four of them after initial Stanford type A and one with chronic type B dissection. The median maximal aortic diameter was 58.0 mm (IQR: 55.0–78.0). Six patients (85.7%) underwent a previous aortic surgery, five an open aortic repair, and one an aortic-subclavian bypass followed by simultaneous TEVAR.

### 3.3. Minimally Invasive Staged Segmental Artery Coil Embolization Characteristics and Outcomes

MIS^2^ACE was successful in 11 sessions (92%) with a median procedural time of 153.0 min (IQR: 116.0–192.0). The median number of embolized arteries was 4 (IQR: 1–4). The distribution of arteries embolized is presented in Figure 1. The median fluoroscopy time and dose–area product were 61.0 min (IQR: 42.0–83.0) and 162.0 Gycm2 (IQR: 118.0–188.0), respectively, while the median contrast medium used during the procedure was 113.0 mL (IQR: 84.0–145.0).

There was no SCI either after MIS^2^ACE or periprocedural complications in any embolization. No patients died, and there was no acute renal injury event at 30 days post-MIS^2^ACE.

The aortic repair followed after a median embolization-operation interval of 37.0 days (IQR: 31.0–78.0). One patient (14%) died at 36 h after simultaneous thoracic and fenestrated endovascular aortic repair due to severe bilateral acute limb ischemia, followed by a compartment syndrome of both lower legs and thighs requiring fasciotomy, finally leading to multiorgan failure and death. This patient had known peripheral artery disease with previous interventions and prolonged endovascular treatment due to multiple technical difficulties and intraoperative rupture of the external iliac artery. We were not able to prove a potential SCI in this patient. Two other patients had open thoracoabdominal and four complex endovascular aortic repair. There was no SCI within 48 h after the subsequent aortic repair. None of the patients in this series had CSFD perioperatively.

One patient with connective tissue disease, who had a complex endovascular repair and was on anticoagulation due to a mechanical aortic valve, was readmitted within one week because of paraparesis and sphincter dysfunction. An urgent MRI demonstrated no SCI but intraspinal bleeding of a known Tarlov cyst between L3 and S3 and was treated conservatively. The neurologic deficits regressed postoperatively with the restoration of sphincter and lower-limb function that remained, albeit reduced, causing a reduction in overall mobility after six months postoperatively.

### 3.4. Meta-Analysis

Figure 2 shows the flow diagram for study selection. Out of the 432 initially retrieved articles, we included two studies in the meta-analysis, including patients with MIS^2^ACE for spinal cord preconditioning in addition to our cohort. Table 2 shows the baseline characteristics and the perioperative/early postoperative MIS^2^ACE-related outcomes of the cohorts included in the meta-analysis. Results regarding the risk of bias assessment of the included studies are summarized in Table 3. Figure 3 shows that the prevalence of pooled postoperative spinal cord ischemia among MIS^2^ACE patients is 1.9% (95% CI −0.028 to 0.066, *p* = 0.279; 3 studies; 81 patients, 127 coiling sessions).

## 4. Discussion

Segmental arteries occlusion with coil embolization prior to open or endovascular aortic repair in our patient cohort was safe and successful. We did not observe any SCI either after MIS^2^ACE or after the aortic repair. While there were no complications after MIS^2^ACE, we had one death and one severe complication after the aortic repair. The patient who died because of intraoperative technical difficulties followed by acute bilateral leg ischemia and consequent multiorgan failure could not be proven for any SCI signs. The other patient with spontaneous bleeding in the Tarlov cyst under anticoagulation and without CSFD, after being discharged without any neurological symptoms, had no signs of CSI in the MRI one week after the aortic repair. Early postoperatively, Branzan et al. reported three deaths (5%, 3/57), and Addas et al. one death (6%, 1/17), the latter one after suffering early postoperative paraplegia [4,8]. Neither series reported any periprocedural complications after MIS^2^ACE [4,8]. Although limited in numbers, our meta-analysis shows a clear tendency that MIS^2^ACE is a safe procedure with low morbidity and mortality. With its staged, multiple sessions approach, it might add an additional tool to the preoperative strategy of reducing the risks for perioperative SCI in TAAA treatment. A pooled postoperative SCI prevalence among MIS^2^ACE patients of 1.9% is significantly lower than any previously reported risk for SCI, especially when the vast majority of the meta-analysis patients had type II or III TAAA extent [1,2]. This low risk for SCI seems to be independent of the embolization-operation time interval and the type of aortic repair. All three studies in the meta-analysis had different embolization-operation time intervals of 37, 51, and 83 days, respectively. We tend to treat the underlying aortic pathology in our patients one month after MIS^2^ACE, whereas, in Leipzig, the tendency was almost three months after the last MIS^2^ACE. The recommended time interval between two MIS^2^ACE sessions and the aortic repair after the last MIS^2^ACE session is five days, but this is based on tests in the porcine model and has to be confirmed in humans [9]. Similar to the treatment algorithm in Leipzig, we tend to repeat as many MIS^2^ACE sessions as needed with at least a seven-day break between two sessions, and we report a median of four occluded segmental arteries per session. Branzan et al. reported a median of five occluded segmental arteries per session, with only 40% of all patients having one session, and the rest had at least two coiling sessions [4]. Addas et al. conducted only one MIS^2^ACE session per patient with a median of three occluded segmental arteries per patient [8]. After the aortic treatment, they reported two MIS^2^ACE patients with paraparesis and one patient with paraplegia after unsuccessful embolization [8]. In this series, all patients had perioperative CSFD and still encountered a higher incidence of SCI [8]. This might ignite the discussion of the benefit of CSFD on one side and the minimum number of occluded segmental arteries on the other side, including the meaning of the occlusion pattern prior to the aortic repair. In the porcine model, von Aspern et al. demonstrated less SCI and favorable neurological outcomes for the MIS^2^ACE staged approach vs. 1-stage occlusion [10]. Additionally, a regional-based occlusion pattern, starting with the lumbar segmental arteries, seems to be the best 2-stage approach [10]. Although with similar results, the included three studies in this meta-analysis have differences concerning the segmental arteries occlusion strategy. Creating a clearly defined coil embolization protocol with a number of sessions needed, identification of the target segmental arteries, MIS^2^ACE under local vs. general anesthesia, etc., could add success to this approach. Performing a segmental arteries occlusion under local anesthesia allows for maximal neurological observation in awake patients. However, coil embolization of segmental arteries, especially the intercostal ones in a large aneurysm or in both lumina of a dissected aorta, can be technically very demanding, requiring an increased amount of contrast medium, radiation, and longer procedure time with potential patient’s malcompliance. The learning curve might play an additional role, especially at the beginning. Out of twelve sessions we performed, the first one was unsuccessful. Similarly, Addas et al. reported three unsuccessful sessions out of 17. With a higher number of procedures, like in the series of Branzan et al., the technical success rate increased over time.

Despite some promising results, there is limited experience to support MIS^2^ACE as a novel technique. The long-expected multicenter, multinational, randomized controlled trial, PAPAartis trial, announced its start in 2019. It planned to include 500 patients worldwide, randomizing them in two groups (MIS^2^ACE vs. no preoperative segmental arteries embolization) in a 1:1 ratio [7]. In the MIS^2^ACE arm, a lower rate of SCI was expected 30 days after aneurysm repair. Due to recruitment difficulties, the trial has yet to be completed, and the final results are eagerly awaited. Until then, MIS^2^ACE remains a valid alternative to reduce postoperative SCI incidence in high-volume aortic centers with sufficient expertise in segmental arteries occlusion.

**Table 3 jcm-13-01408-t003:** Risk of bias assessment *.

Study	Q1	Q2	Q3	Q4	Q5	Q6	Q7	Q8
Branzan et al., 2018	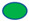	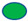	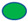	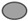	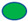	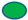	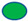	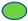
Addas et al., 2022	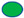	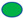	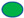	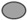	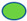	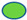	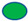	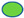
Dabravolskaitė et al., 2023	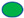	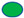	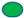	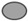	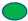	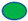	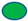	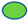

Q = Question; 
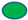
 = Definitely Yes; 
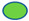
 = Probably Yes; 
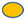
 = Probably No; 
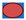
 = Definitely No; 
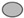
 = Not applicable; D = Domain; 
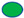
 = Low; 
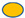
 = Some concerns; 
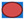
 = High. * Assessed by the “Tool to assess risk of bias in cohort studies” by the CLARITY Group at McMaster University (https://www.distillersr.com/resources/methodological-resources/tool-to-assess-risk-of-bias-in-cohort-studies-distillersr) [11] modified for single-arm studies (accessed on 27 November 2023).

## 5. Conclusions

MIS^2^ACE is a safe method with a low risk of periprocedural complications and a lower rate of spinal cord ischemia during open or endovascular aortic repair for thoracoabdominal aortic aneurysm. This meta-analysis favors the use of a planned, multiple, staged segmental arteries occlusion approach in minimizing the early postoperative risk for spinal cord ischemia after TAAA repair.

## Figures and Tables

**Figure 1 jcm-13-01408-f001:**
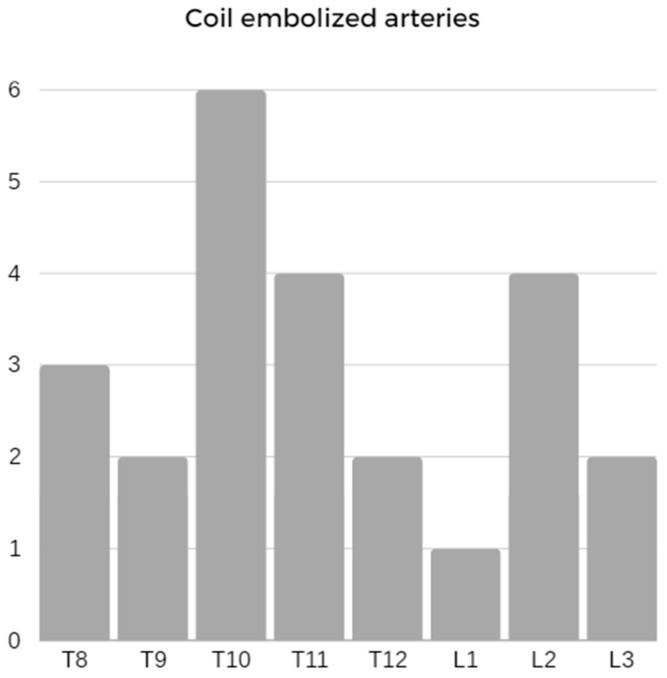
Distribution of coil-embolized segmental arteries.

**Figure 2 jcm-13-01408-f002:**
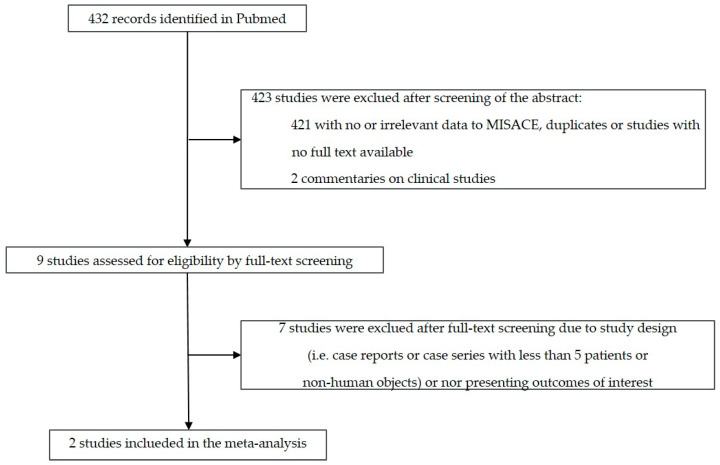
Flow diagram for study selection.

**Figure 3 jcm-13-01408-f003:**
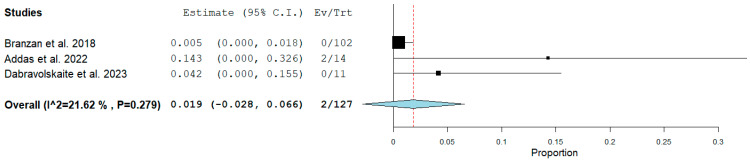
Prevalence of pooled postoperative spinal cord ischemia among MIS^2^ACE patients (Forest plot).

**Table 1 jcm-13-01408-t001:** Baseline characteristics and outcome of patients undergoing minimally invasive staged segmental artery coil embolization.

Number of patients, *n*	7
Number of total sessions, *n*	12
Age, years	57.0 (55.0–69.0)
Female sex, %	2 (28.6)
Underlying aortic pathology	
Crawford Classification	
extent I	0 (0.0)
extent II	5 (71.4)
extent III	2 (28.6)
extent IV	0 (0.0)
Stanford Classification	
type A	4 (80.0)
type B	1 (20.0)
Degenerative aneurysm	2 (28.6)
Post-dissection aneurysm	5 (71.4)
Previous aortic repair, *n*	6 (85.7)
Maximal aortic diameter, mm	58 (55–58)
Embolization-operation interval, days	37 (31–78)
Arteries coiled, *n*	4 (1–4)
Procedure time, min	153 (116–192)
Fluoroscopy time, min	61 (42–83)
Contrast medium used, mL	113 (84–145)
Fluoroscopy dose–area product, Gycm^2^	162 (118–188)
Unsuccessful MISACE session, *n*	1 (8)
Periprocedural major bleeding, *n*	0 (0.0)
Endoleak at 30 days, *n*	0 (0.0)
Spinal cord ischemia at 48 h after MISACE, *n*	0 (0.0)
Spinal cord ischemia at 48 h after aortic repair, *n*	0 (0.0)
Death at 30 days after MISACE, *n*	0 (0.0)
Death at 30 days after aortic repair, *n*	1 (14)

Abbreviations: *n*, number; mm, millimeter; min, minutes; mL, milliliter. Data are presented as median (interquartile range) or number (%).

**Table 2 jcm-13-01408-t002:** Characteristics and perioperative/early postoperative MIS^2^ACE outcomes of included cohorts.

Study	Type of Study	Country	Total Number of Patients, *n*	Total number of MISACE Sessions, *n*	Age (Years)	Female (%)	Crawford Classification	Mean Aneurysm Size, mm	Technically Successful MISACE, %	Embolized Arteries, *n*	Perioperative Complications, %	Spinal Cord Ischemia, %
Branzan et al., 2018	Single-center retrospective observational	Germany	57	102	69.6 ± 7.6	25	Type I, 8.8% Type II, 21.1% Type III, 47.3% Type IV, 22.8%	62.7 ± 8.8	99.7	5 (1–19)	13.7	0
Addas et al., 2022	Single-center retrospective observational	Canada	17	17	69.0 (47.0–85.0)	23.5	Type I, 5.9% Type II, 35.3% Type III, 23.5% Type IV, 29.4% Type V, 5.9%	70.6 ± 10.9	82.4	3 (1–6)	0	14
Dabravolskaitė et al., 2023	Single-center retrospective observational	Switzerland	7	12	57.0 (55.0–69.0)	28.6	Type I, 0.0% Type II, 71.4% Type III, 28.6% Type IV, 0.0%	58.0 (55.0–58.0)	91.7	4 (1–4)	0	0

Abbreviations: *n*, number; mm, millimeter; MIS^2^ACE, Minimally Invasive Segmental Artery Coil Embolization. Data are presented as median with interquartile range, mean with standard deviation, or number (%).

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
