# Peer review of "The Safety and Outcome of Minimally Invasive Staged Segmental Artery Coil Embolization (MIS2ACE) Prior Thoracoabdominal Aortic Aneurysm Repair: A Single-Center Study, Systematic Review, and Meta-Analysis"

_jcm, 2024, doi:10.3390/jcm13051408_

Round 1
Reviewer 1 Report
Comments and Suggestions for Authors
Thank you for a well written and clearly presented paper. The issue I have with the manuscript as submitted is that it confuses two study types, a retrospective observational study - 7 patients undergoing MIS2 ACE (Bern, Switzerland) - and a systematic review - 432 publications screened / 2 included studies (74 patients).
Branzan (n=57) and Abbas (n=17) only reported on endovascular repair of TAAA following segmental llumbar coil embolisation.
Methodologically, it is incorrect to include unpublished data (this study) alongside the two published studies when analysing the meta-analysis (Table 2).
The Bern study includes both open and endovascular case, open and endovascular repair have different rates of paraplegia and cases cannot be combined in this way.
Other points:
1. No description of systemic review findings in the publication abstract.
2. PAPAartis trial name misspelled in manuscript.
3. Page 9, Line 21 - There is no evidence presented that MIS2ACE is valid, as no control group is present. This statement cannot therefore be supported. What has been presented is safety data. (This sentence also contradicts the statement on Page 10, Line 60 'not enough literature to support'.
Author Response
Reviewer 1.
Thank you for a well written and clearly presented paper.
The issue I have with the manuscript as submitted is that it confuses two study types, a retrospective observational study - 7 patients undergoing MIS2 ACE (Bern, Switzerland) - and a systematic review - 432 publications screened / 2 included studies (74 patients).Branzan (n=57) and Abbas (n=17) only reported on endovascular repair of TAAA following segmental llumbar coil embolisation. Methodologically, it is incorrect to include unpublished data (this study) alongside the two published studies when analysing the meta-analysis (Table 2).
Thank you for the comment. It is important to critically evaluate the quality of the unpublished data when considering its inclusion to the meta-analysis since unpublished studies may not have undergone the peer review process the same way published studies. Nevertheless, based on various literature sources, included study by Maria J. Grantet al ( https://onlinelibrary.wiley.com/doi/10.1111/j.1471-1842.2009.00848.x) , meta-analyses can include unpublished data. In fact, incorporating unpublished studies in a meta-analysis can be important for several reasons:
- It may reduce the publication bias.
- Publishing comprehensive studies ensure aggregation of data relevant to the research question.
- It increases statistical power.
With your permission we would like to keep the current manuscript design and present our observational study as a part of the systematic review and meta-analysis.
The Bern study includes both open and endovascular case, open and endovascular repair have different rates of paraplegia and cases cannot be combined in this way.
Thank you for the comment. We are truly aware of the differences in paraplegia rates after open or endovascular TAAA treatment. However, our work focused primarily on the safety and the outcome of MIS2ACE itself, as a procedure. Additionally, we presented the outcome of the following TAAA treatment, concerning the transitory or permanent spinal cord ischemia, thus being in line with the other literature from the systematic review. We believe that this adds a valid information for the readers, independently of the treatment strategy.
Other points:
- No description of systemic review findings in the publication abstract.
Thank you for this remark. We added now the most relevant information from the systematic review and meta-analysis in the abstract as suggested. - PAPAartis trial name misspelled in manuscript.
Thank you for this remark, we corrected this in the manuscript. - Page 10, Line 21 - There is no evidence presented that MIS2ACE is valid, as no control group is present. This statement cannot therefore be supported. What has been presented is safety data. (This sentence also contradicts the statement on Page 10, Line 60 'not enough literature to support'. We agree with your remark and rephrased this statement, both on Page 10, lines 20-22 “Although limited in numbers, our meta-analysis shows a clear tendency that MIS2ACE is safe procedure with low morbidity and mortality. With its staged, multiple sessions approach, it might add an additional tool in the preoperative strategy of reducing the risks for perioperative SCI in TAAA treatment” and on Page 11, lines 62 “there is limited experience…”
Reviewer 2 Report
Comments and Suggestions for Authors
Dear Authors,
I would like to thank you for your interesting study. Although the number of patients is small, which is the main limitation of your study, it provides a clear message of the benefits of MISACE technique in aortic pathology.
The only thing you should add in your manuscript is a more detailed operational technique of embolization of intercostal arteries.
Author Response
Reviewer 2.
Dear Authors,
I would like to thank you for your interesting study. Although the number of patients is small, which is the main limitation of your study, it provides a clear message of the benefits of MISACE technique in aortic pathology.
The only thing you should add in your manuscript is a more detailed operational technique of embolization of intercostal arteries.
Thank you for pointing this out. The technique of the procedure is now added to the manuscript on pages 2 and 3.